# Designing pH-Dependent Systems Based on Nanoscale Calcium Carbonate for the Delivery of an Antitumor Drug

**DOI:** 10.3390/nano11112794

**Published:** 2021-10-21

**Authors:** Victoriya Popova, Yuliya Poletaeva, Inna Pyshnaya, Dmitrii Pyshnyi, Elena Dmitrienko

**Affiliations:** Institute of Chemical Biology and Fundamental Medicine (ICBFM), Siberian Branch of Russian Academy of Sciences (SB RAS), 8 Lavrentiev Avenue, 630090 Novosibirsk, Russia; fom.nin198@mail.ru (V.P.); fabaceae@yandex.ru (Y.P.); pyshnaya@niboch.nsc.ru (I.P.); pyshnyi@niboch.nsc.ru (D.P.)

**Keywords:** calcium carbonate nanoparticle, *DOX*orubicin, prolonged release

## Abstract

Materials based on calcium carbonate (CaCO_3_) are widely used in biomedical research (e.g., as carriers of bioactive substances). The biocompatibility of CaCO_3_ and dependence of its stability on pH make these materials promising transporters of therapeutic agents to sites with low pH such as a tumor tissue. In this work, we developed an approach to the preparation of nanoscale particles based on CaCO_3_ (CaNPs) up to 200 nm in size by coprecipitation and analyzed the interaction of the nanoparticles with an anticancer drug: *DOX*orubicin (*DOX*). We also showed a prolonged pH-dependent release of *DOX* from a CaNP nanocarrier and effective inhibition of cancer cell growth by a CaCO_3_-and-*DOX*–based composite (CaNP_7_-*DOX*) in in vitro models.

## 1. Introduction

Calcium carbonate (CaCO_3_) is one of the most common inorganic materials with a wide range of applications in biomedical fields, for example, as the basis for delivery systems of biologically active substances or for biosensor construction [1]. Studies on the use of CaCO_3_ particles in biomedicine date back to the 1990s. The advantages of CaCO_3_-based materials include good availability of reagents, the absence of toxicity, and the gradual biodegradation of CaCO_3_ nanoparticles (CaNPs) [2]. The biocompatibility of CaNPs toward cells has also been reported, as has the suitability of such materials as safe drug carriers [3].

A large number of studies have been published aimed at the design of Ca^2+^-based delivery systems for anticancer drugs (including *DOX*orubicin; *DOX*) [4,5]. Most of the studies on the preparation and characterization of nanomaterials based on CaCO_3_ involve porous particles in the micrometer size; the advantages of such materials include the large surface area (due to the porosity of such structures) available for interaction with the drug; however, the large particle size imposes significant limitations in the use of such materials in vivo [6,7]. The literature indicates that for the development of cancer drug delivery vehicles, promising strategies are those that combine both an active targeting modality (for example, transferrin) and a passive one (i.e., a targeting system based on the enhanced permeability and retention effect in deformed cancerous tissues). In the case of passive targeting, nanomaterials with sizes <200 nm are optimal; such nanoparticles have a better ability to penetrate into (and accumulate at) the tumor site due to defects in endothelial cells [8].

There are significantly fewer scientific publications about the preparation of CaNP materials with sizes less than 200 nm than publications about their micrometer scale analogs because of difficulties (i) with the development of the methods for constructing such materials and (ii) with their subsequent stabilization. Zhao Y et al. presented a technique for the fabrication of nanoscale monodisperse CaNPs, but the material was not stable in an aqueous solution, therefore, it was functionalized with a silicate shell for further applications. This modification significantly lowered the efficiency of drug encapsulation and reduced the ability of *DOX* to desorb from the CaNP-based particles, namely, those authors failed to achieve a release of more than 25% of the loaded drug at a physiological temperature [9]. Hamidu A. et al. conducted a lot of work on the preparation and characterization of CaNPs loaded with *DOX*, but were unable to prevent aggregation of the nanomaterials into macro objects [10]. Wenliang Fu and colleagues [11] worked with similar nanomaterials. The poor monodispersity significantly limits prospective applicability of CaNPs to biomedicine due to a substantial increase in the size of the particles because they tend to adhere to each other.

Despite the current absence of FDA-approved CaCO_3_-based delivery vehicles (carriers) of cancer drugs for clinical use, this material undoubtedly holds promise. Scientists continue to develop protocols of CaNP construction for subsequent biomedical applications including in vivo experiments [12,13].

Wenliang F. et al. demonstrated the potential utility of CaNPs (a size distribution of 20–60 nm; prepared from seashells) as a drug delivery system for the treatment of osteosarcoma in an orthotopic rat model of this cancer. After measuring body weight and analyzing serum biochemical parameters and histopathological data, the authors showed that *DOX* administration caused its accumulation (and the manifestation of toxic side effects) in major organs such as the heart, liver, and kidneys. In contrast, during CaCO_3_-*DOX* therapy, the manifestation of the adverse effects was weaker and the survival rate of the rats was higher compared to the rat group receiving free *DOX*. These researchers argued that this benefit was due to increased passive targeting to tumor tissues because of the enhanced permeability and retention effect (EPR) and pH sensitivity of CaCO_3_, namely, that this sensitivity reduced systemic toxicity and unintended exposure of normal tissues to *DOX* while increasing the chemotherapeutic action on solid tumors [14]. Maleki Dizaj S. and colleagues investigated the efficacy of *DOX*-loaded pH-dependent CaNPs (<600 nm) in dogs with tumors including osteosarcoma in a single-center open-label clinical trial; they registered osteoid matrix production, death of osteogenic cells, and inhibition of their proliferation [4]. Ghaji M. S. et al. studied CaNPs (20–50 nm) as a carrier of cytarabine acting against human leukemic HL-60 cells and as an antitumor therapy in SCID mice with leukemia; half-maximal inhibitory concentrations (IC_50_) of cytarabine and CaNP-cytarabine were 5 and 2.5 μg/mL, respectively, after 72 h. CaNP-cytarabine was more effective at inducing apoptosis than free cytarabine [15]. Nonetheless, due to a high polydispersity index (PDI), heterogeneity and aggregation of CaNPs, protocols for the synthesis of such nanomaterials require modification. Nanoparticles can be entrapped in some organs and tissues, this problem is relevant for all nanoparticles. However, the advantages of nanoparticles based on calcium salts are their non-toxicity and biodegradability, which are confirmed by studies in the literature [16]. Moreover, for calcium phosphate nanoparticles of a submicron size, the distribution over organs and the subsequent excretion including through the decomposition of nanoparticles has been shown [17]. Thus, we expect that the biodegradability of nanoparticles based on calcium carbonate will avoid the negative effects of the accumulation of nanoparticles in organs.

Natural properties of CaCO_3_-based materials such as biocompatibility, biodegradability, and pH sensitivity make CaCO_3_ a promising candidate carrier for the delivery of various biologically active substances, especially anticancer drugs [18,19]. In the present work, we focused on the materials science part of this topic to develop a fundamental approach to the fabrication of nanomaterials with the special characteristics that will increase the efficacy of CaCO_3_ applications in biomedicine.

CaCO_3_ particles that meet several criteria—monodispersity, stability under physiological conditions, partial or complete degradation at low pH levels, and size <200 nm—are attractive carriers for pH-dependent drug delivery [20]. In the literature, there are several basic approaches to the preparation of CaCO_3_ particles. Depending on the synthesis technique, it is possible to obtain particles of various shapes and sizes. Options for constructing CaNPs are subdivided into three main categories: aeration or carbon dioxide barbotage, coprecipitation of appropriate salts, and microemulsion approaches [21]. The easiest option to implement is the coprecipitation method because it does not require special equipment and reagents. Despite the apparent simplicity of the existing techniques and numerous studies aimed at obtaining CaNPs [22,23,24], there is no simple and reproducible way to synthesize monodisperse CaCO_3_ particles of the nanometer size (up to 200 nm) that preserves their stability in solution for prospective in vivo applications. Thus, the main aim of this study was to find a way to prepare a suspension of nanoscale CaNPs including complete screening of various synthesis conditions and assessing the effects of reaction mixture composition, solvents, pH, and various additives on the morphological properties of the formed particles, with subsequent testing of the interaction of the particles with a drug (*DOX*). Additionally, we aimed to study the efficiency of conjugation of CaNPs with the drug, the kinetic release profile of *DOX* from the matrix (CaNPs) depending on pH, and the effectiveness of the inhibition of cancer cell proliferation by CaNP carrying *DOX* in vitro.

In this work, we developed and described a simple approach to the fabrication of nanocomposites based on CaCO_3_ and researched the physical and chemical properties of these nanoparticles. We noted the absence of toxicity and proved the effectiveness of CaNPs as a drug carrier.

## 2. Materials and Methods

### 2.1. Materials

Sodium hydrogen carbonate, calcium chloride, magnesium chloride, and Tween 20 were purchased from Sigma-Aldrich, Co (St. Louis, MO, USA) whereas *DOX* from Teva Pharmaceutical Industries Ltd. (Petach Tikva, Israel). 3-(4,5-Dimethyl-2-thiazolyl)-2,5-diphenyl-2H-tetrazolium bromide (MTT) was acquired from Panreac Química (Barcelona, Spain), and polyethylene glycol (PEG) 1000 Da, 2000 Da, and 6000 Da from Carl Roth (Karlsruhe, Germany), FBS (fetal bovine serum), DMEM (Dulbecco’s modified Eagle medium), and antimycotic antibiotic solution from GIBCO, Life Technologies (Carlsbad, CA, USA).

### 2.2. Synthesis of CaNPs by Coprecipitation

This procedure was performed by mixing two salts on an ultrasonic bath or dispersant: 100 µL of a CaCl_2_ solution (0.007–0.100 M) was added dropwise to 1 mL of a NaHCO_3_ (0.1 M) aqueous solution in the absence and presence of additives such as PEG (MW = 1000, 2000, or 6000) at 0.1 mg/mL, detergents (Tween 20, Triton X-100, sodium dodecyl sulfate, or cetyltrimethylammonium bromide) at 0.1 vol.%, DMEM at 0.2–10.0 vol.%, and/or MgCl_2_ (0.005–0.010 M) [25]. The additives were introduced both individually and as a mixture. The reaction of CaNP formation in a mixture of Tween 20 and PEG-2000 was performed similarly to the reaction in an aqueous solvent replaced by isopropanol [1].

### 2.3. Characterization of CaNPs

This was carried out by dynamic light scattering (DLS) methods on a Malvern Zetasizer nano particle characterization system in water at room temperature. Suspensions of CaNPs were analyzed by transmission electron microscopy (TEM). For this purpose, a drop of a sample was allowed to adsorb for 1 min on a copper grid covered with formvar film; the excess liquid was then removed, and the grids were examined under a Jem1400 microscope (Jeol, Tokyo, Japan). Images were captured by a side-mounted Veleta digital camera (EM SIS, Muenster, Germany).

### 2.4. Reversible Binding of CaNPs to DOX

Conjugation was performed with stirring (700 rpm, 25 °C, 12 h) in 10 mM borate buffer (pH 8.5) containing 0.1–1.0 mg/mL CaNPs and 50–500 µg/mL *DOX*. Next, the CaNPs were washed with 10 mM borate buffer pH 8.5 (3 times × 1 mL), and the supernatant was separated by centrifugation (10 min, 13,400 rpm, miniSpin from Eppendorf). The residual concentration of *DOX* in the supernatant was determined spectrophotometrically in the wavelength range of 400–600 nm at room temperature.

The amount of the drug bound to the nanoparticles was determined as capacity (E) of CaNPs for the drug and calculated according to the formula:$$E=\frac{DOX_{0}-DOX}{N}$$
where E is particle capacity (µg *DOX*/mg CaNPs); *DOX*_0_ represents the initial amount of *DOX* (µg); *DOX* is the amount of *DOX* in the supernatant (µg); and *N* denotes the amount of CaNPs (mg).

The release of *DOX* was investigated at room temperature (unless stated otherwise; in some cases, the temperature was varied from 15 to 45 °C) in 1 mL of 100 mM acetate buffer (pH of 3.0 to 7.0) containing CaNP-*DOX* (132 μg of *DOX* with 0.2 mg of CaNP_7_) with constant stirring (750 rpm). The amount of *DOX* released into the solution was determined by means of optical density and/or fluorescence intensity of the solution.

### 2.5. The Cytotoxicity of CaNPs, CaNP-DOX, and DOX

These properties were evaluated on A549 and HEK293 cells in a standard MTT assay. For this purpose, cells at (2.0 ± 0.5) × 10^3^ per well were seeded in a 96-well plate containing the culture medium and incubated for 24 h at 37.0 ± 1.0 °C in an atmosphere containing 5.0% ± 0.5% of CO_2_. HEK293 and/or A549 cells were then incubated with CaNPs (0.2–22.5 mg/mL), CaNPs carrying *DOX* (0.1–10.0 μM), or free *DOX* (0.1–10.0 μM) in the culture medium for 48 h at 37.0 ± 1.0 °C in an atmosphere containing 5.0% ± 0.5% of CO_2_. After that, the medium was removed from all wells, and 200 µL of an MTT solution (0.25 mg/mL in the culture medium containing 1% of an antimycotic antibiotic solution) was added and incubated for 4 h under the same conditions. Next, the medium and MTT were removed from the wells, and 100 µL of dimethyl sulfoxide (DMSO) was added, and optical density was measured on a multichannel plate reader at wavelengths of 570 and 450 nm. The percentage of surviving cells was calculated from the obtained optical density values for each concentration of a tested agent.

### 2.6. Statistical Evaluation of Experimental Error

This was performed on at least three parallel biological samples. Data obtained in each experimental series are presented as mean ± standard deviation calculated by Excel software.

## 3. Results and Discussion

### 3.1. Synthesis and Assessment of CaCO_3_ Properties

The synthesis of CaNPs by coprecipitation is a simple, inexpensive, and efficient method for producing nanomaterials for biomedical applications. On the other hand, micrometer scale particles (DLS data: 2950 ± 400 nm hydrodynamic diameter [d], PDI = 0.1 ± 0.05) are obtained by equimolar mixing of CaCl_2_ and NaHCO_3_ in the absence of additional reagents and do not meet our criteria. To obtain stable monodisperse particles up to 200 nm, we investigated the influence of the reaction mixture composition on the morphological characteristics of CaNPs. The workflow of CaNP fabrication is shown in Figure 1.

An ultrasonic bath (for small-scale synthesis up to 1.5 mL) and a dispersant (up to 50 mL) were employed in the synthesis setup for all CaNPs.

The effect of stoichiometric ratios of the reagents on the size of the resulting particles was evaluated first. By varying the calcium chloride concentration (0.007–0.100 M) at a constant concentration of the carbonate anion and vice versa, we found that the size of formed particles at all tested concentrations was greater than 1 micron. According to DLS data, the use of a 10-fold excess of CO_3_^2−^ over Ca^2+^ allows particles of the smallest size in the examined range to be obtained (d = 1470 ± 180 nm, PDI = 0.635 ± 0.002). Therefore, the optimal concentrations (0.1 M NaHCO_3_ and 0.01 M CaCl_2_) were chosen for further experiments. The addition of surfactants and high-molecular-weight compounds is widely utilized for preparing nanoparticles (up to 200 nm) [26]. The effects of adding the following surfactants to the reaction mixture were studied here: sodium dodecyl sulfate (anionic detergent), cetyltrimethylammonium bromide (cationic detergent), Tween 20 and Triton X-100 (nonionic detergents), and biocompatible high-molecular-weight compounds (PEG-1000, -2000, and -6000) [26]. According to the DLS data, only the addition of detergent Tween 20 led to a significant reduction in particle size (d = 450 ± 30 nm, PDI = 0.11 ± 0.03). Supplementation with PEG, irrespective of molecular weight (MW), yielded particles larger than 700 nm, which were still smaller than the particle sizes obtained in the absence of additives. Accordingly, the effects of the combined addition of high-molecular-weight PEG (1000, 2000, or 6000) and detergent Tween 20 were investigated next. The combined addition of these compounds reduced the size of CaNPs in comparison with the nanomaterials obtained above by separate supplementation with either compound alone. The smallest hydrodynamic size (d = 340.2 ± 0.3 nm, PDI = 0.177 ± 0.003) of CaNPs at this stage of the work was achieved upon the combined addition of PEG-2000 and Tween 20 to the reaction mixture. To determine the influence of codoping with PEG-2000 and Tween 20 on the CaNP formation process, both additives and their combination were examined by TEM. Figure 2 presents the fine structure of the initial solutions of Tween 20 (Figure 2A) and PEG-2000 (Figure 2B) as well as their combination (Figure 2C).

According to the figure, each individual additive has an ultrastructure that is different from that seen when they are mixed. From the TEM data (Figure 2), it was concluded that PEG-2000 and Tween 20 together formed a polymeric structure that acts as a matrix limiting particle growth during nucleation.

The CaNPs obtained in the presence of PEG-2000 and Tween 20 (called CaNP_1_) were also characterized by TEM (Figure 3).

An analysis of the TEM micrographs (Figure 3) revealed that the obtained CaNPs were heterogeneous in size and shape, and polydispersity was considerable: particle size ranged from 135 to 700 nm.

To increase the stability and monodispersity of the CaNPs and to reduce their size, we then screened other conditions for CaNP fabrication. It is known that the addition of a nutrient medium, DMEM, consisting mainly of amino acids, inorganic salts, and vitamins, and Mg^2+^ cations, to the reaction mixture results in the formation of small CaNPs and in their stabilization [25]. The reason is presumably competing processes and the slowing of crystal formation. Accordingly, we investigated the effects of adding DMEM, MgCl_2_, or their combination—to the reaction mixture consisting of a salt solution with a 10:1 ratio of CO_3_^2−^ to Ca^2+^ and containing PEG-2000 and Tween 20—on the physicochemical properties of the CaNPs formed. Aside from the additives, the influence of the solvent on the characteristics of the obtained CaNPs [1] was also evaluated (Table 1).

All of the obtained samples were stable in solution (hereafter stability is defined as the preservation of the physical and chemical properties of CaNPs) and were analyzed by TEM to examine the fine structure of the materials (Figure 4).

It was obvious that the composition of the reaction mixture during the CaNP synthesis significantly affected the morphological properties of the nanoparticles: size and ultrastructure (Table 1 and Figure 4). As above-mentioned, the presence of PEG-2000 and Tween 20 (CaNP_1_, Figure 4A) yielded reproducible fabrication of CaNPs with a wide size distribution beyond the range optimal for biomedical applications, and this phenomenon is related to the tendency of CaNPs to aggregate. Replacing the aqueous solvent with an organic one caused an increase in crystal growth efficiency, which significantly enlarged the particles (CaNP_2_, Figure 4B). The samples obtained in the presence of Mg^2+^ (CaNP_3_, Figure 4C) were visibly different from those above, according to the DLS and TEM data. This is because the DLS method cannot determine the size of individual particles; instead, it only quantifies the clusters arising via aggregation of the material, as demonstrated in the TEM images. Because it was not possible to overcome the problem of particle aggregation by either chemical or physical methods, these CaNPs were not used in further work, despite their small size. The addition of DMEM in the absence of magnesium chloride did not cause cardinal differences from the sample obtained in the presence of Tween 20 and PEG (Figure 4A,D). In the comparison of panels E–H (Figure 4), the combined supplementation with 10% DMEM and 0.01 M MgCl_2_ (CaNP_7_, Figure 4G) was found to be optimal. Stable monodisperse spherical 249.0 ± 0.8 nm CaNPs were obtained, which were closest to the requirements of the task in question.

Storage and scalability tests of the nanoscale CaNPs fabricated by the newly developed procedure were performed on CaNP_7_. The stability of these CaNPs was investigated by remeasuring the hydrodynamic dimensions of the CaNPs three months after the synthesis. The physical characteristics of the samples did not change much. The scalability of the proposed approach was assessed by increasing the reaction mixture volume by 30-fold, which made it necessary to replace the ultrasonic bath with a dispersant. According to the DLS (d = 200 ± 20 nm, PDI = 0.10 ± 0.04) and TEM data (Figure 5), the larger synthesis volume did not increase the hydrodynamic size of the CaNPs.

This nanomaterial was also assayed for stability. During storage for three months, the hydrodynamic size did not change significantly (Figure 6).

As discussed elsewhere, obtaining stable CaNPs in an aqueous solution without additional modifications of the procedure is not an easy task [9], and an aqueous solution is a prerequisite for subsequent successful application of the nanomaterials. The stability of our CaNPs in aqueous solutions, the scalability of our synthesis methodology, and good availability of the reagents will allow for the convenient use of these nanomaterials in the chemical and pharmaceutical industries: the synthesized nanomaterial called CaNP_7_ can be stored while retaining its original properties for >3 months.

To evaluate the characteristics and stability of CaNP_7_ in the bloodstream, we conducted a model experiment on the storage of the nanoparticles in a 50% serum solution (Figure 7).

The increase in the hydrodynamic particle size, in comparison with the initial one (t = 0 and t = 5 min, Figure 7), can be primarily explained by the change in the solution in which the DLS analysis was performed; this alteration can affect the obtained hydrodynamic radius. Because the particles between the time points of 5 min and nine days retained their size within the margin of error, it can be said that their stability in serum is sufficient for prospective biomedical applications [27].

Accordingly, the impact of solution composition (addition of a surfactant, DMEM, MgCl_2_, and a change of the solvent) on the characteristics of the obtained materials was investigated experimentally based on the literature data. As a result, several simple and accessible approaches to obtaining monodisperse CaNPs of different shapes (spherical and rod-shaped) and sizes (from 40 to 200 nm) were developed here. Then, a methodology yielding CaNPs suitable for further biomedical research (CaNP_7_) was selected. The obtained nanomaterial was found to have the parameters necessary for further investigation of the potential usefulness of CaNPs as a carrier of model antitumor agents.

### 3.2. Interaction of DOX with CaNP_7_

There are three techniques for conjugating a biologically active compound with a carrier. Water-soluble therapeutic agents can be coprecipitated during nanomaterial synthesis [28]. Another approach is based on the impregnation of the prepared particles with a solution of the drug with constant stirring or shaking [29]. In this case, the binding of the obtained nanoparticles to the drug can be achieved either via adsorption or encapsulation. The third option for the conjugation of the nanomaterial with the drug is based on evaporation of a solvent containing the drug under reduced pressure. The latter two methods are suitable for our purpose because they can load drugs that are poorly soluble in water, aside from other advantages. In our study, we chose the method of postsynthetic impregnation of CaNPs with the drug.

It is likely that the efficiency of adsorption of macromolecules inside CaCO_3_ particles or onto their surface is determined by electrostatic interactions. In heterogeneous systems (solid–liquid), the relevant factors are steric effects, molecular weight of the drug being immobilized, and its affinity for the CaNP surface. Via adjustment of pH in the reaction mixture, it is possible to change electrostatic interactions and, as a consequence, to manage the adsorption–desorption of the drug by controlling the number of its released/loaded molecules [30].

For example, drug-loaded CaNPs are applicable to the controlled release and delivery of various biomolecules.

*DOX* is an antibiotic that is also a common broad-spectrum antitumor agent used in chemotherapy. *DOX* was chosen here as the model therapeutic agent for the evaluation of CaNPs as drug carriers. The disadvantages of this drug are several serious adverse effects, ineffective penetration through the cell membrane, rapid excretion from the body, and poor water solubility due to hydrophobicity [31]. The development of a selective prolonged delivery system for a given drug is a highly relevant task. Maximal absorption of *DOX* in the visible spectral region (480 nm) enables quantitative analysis of the effectiveness of *DOX* conjugation with CaNPs by spectrophotometry without the introduction of additional labels.

The efficiency of CaNP_7_ conjugation with *DOX* was evaluated by means of the drug capacity index, which was calculated as the amount of *DOX* (in μg) bound to 1 mg of CaNP_7_. The amount of the drug bound to the nanoparticles was calculated as the difference between added *DOX* and *DOX* remaining in solution (supernatant) after incubation with CaNP_7_. The CaNP_7_ capacity index was found to be 659 ± 5 μg/mg. On the basis of the literature data, we believe that the nanocomposites obtained in this work have good prospects for practical applications because they are more than threefold superior to most analogs in terms of *DOX* encapsulation efficiency [7,32].

One of the advantages of materials based on CaCO_3_ is pH-dependent stability of the particles: with a decrease in pH, the rate of hydrolysis of the nanocarrier matrix increases, thereby facilitating the release of the encapsulated drug. In the next step, we investigated this phenomenon.

The drug release efficiency of the CaNP_7_–*DOX* nanocomposite as a function of medium acidity was assessed in 100 mM acetate buffer at pH from 3.0 to 7.0. To study the release of *DOX* from the CaNP_7_–*DOX* complex, the amounts of the components (CaNP_7_ and *DOX*) were identical among the assays with different pH levels: 132 μg of *DOX* per 0.2 mg of CaNP_7_.

CaCO_3_ is considered unstable under acidic conditions; a similar property was expected in CaNPs. Figure 8 shows a pH-dependent release of the drug: as pH of the medium diminished, the proportion of released *DOX* increased, partly due to complete or partial degradation of the nanocarrier matrix. At pH 3.0, complete liberation of loaded *DOX* was achieved, whereas at pH close to physiological, the degree of *DOX* liberation was the lowest: at pH 7.0, the drug liberation rate was 25% after 180 min. Therefore, the pH-dependent release of *DOX* from its complex with CaNP_7_ was demonstrated successfully.

High efficiency of a drug release at endosomal pH has been reported elsewhere [33]. In the present study, at physiological pH values, CaNP_7_ retained more than 75% of the loaded drug within the nanocomposite, and this property may ensure high selectivity of drug distribution in the context of a site-activated therapy.

It is worth mentioning that the nanomaterials developed by us here had a high release efficiency of >80% when pH was lowered. This effect can help to implement passive targeting of the drug to tumor tissues, which are known to have more acidic pH than that of healthy tissues [34]. We expect that in future applications, the sensitivity to the weakly acidic pH of the tumor microenvironment will ensure effective highly specific delivery and a sustained release of anticancer drugs in in vivo experiments.

Next, at pH values close to physiological (pH 6.5), where only low drug release efficiency was seen (Figure 8), we examined the impact of temperature (15, 25, 37, and 45 °C) on the release efficiency of *DOX* from its complex with CaNP_7_ (Figure 9).

The efficiency of *DOX* liberation increased with the increasing temperature: the difference between 45 and 15 °C was ~7% at each data point in the experiment (Figure 9).

Thus, it was successfully demonstrated that the obtained CaCO_3_-based materials can form the basis of drug delivery systems. CaNP_7_ effectively binds to the model therapeutic agent. During our investigation into the kinetic profiles of the *DOX* release from CaNP_7_-*DOX*, it was found that CaNPs are pH-sensitive: the drug liberation increases with decreasing pH. This is a promising property for drug delivery to low-acidity sites including malignantly transformed tissues. When the temperature was raised from 15 to 45 °C, the drug release from the tested CaNPs increased. These results suggest that CaNP_7_ can be further investigated as a nanocarrier for drug delivery. On the other hand, the main prerequisite for the use of CaNPs in vivo is the absence of toxicity.

### 3.3. Cytotoxicity Assays of CaNP_7_ and Its Composites with DOX

The cytotoxicity of the obtained CaNPs and nanocomposites before and after complexation with *DOX* (CaNP_7_-*DOX*) was assessed in the standard MTT assay, which estimates the percentage of surviving cells after exposure of the cells to the agent being tested. To evaluate the effectiveness of cell growth inhibition by the CaNP_7_–*DOX* nanocomposite, a comparison was made with *DOX* alone and CaNPs alone in an assay involving a lung carcinoma cell line (A549). The absence of toxicity of CaNPs was successfully proven in an assay involving a human embryonic kidney 293 cell line (HEK293) and A549: at CaNP concentrations up to 22.5 μg/mL, the viability of the treated cells did not diminish below 98%. The literature data confirm the safety of CaCO_3_ micro- and nanoparticles according to MTT assays [6,35]. Nonetheless, we needed to confirm this for the nanomaterial prepared by our method: CaNP_7_.

Figure 10 indicates the effective inhibition of the growth of cancer cells (A549) by the CaCO_3_-based nanocomposite containing an antitumor agent (*DOX*) compared to the drug alone.

The CaNP_7_–*DOX* composite proved to be comparable in efficacy to free *DOX*. This was confirmed by determining IC_50_: the concentration of the inhibitory agent (either CaNP_7_-*DOX* or *DOX*) required to reduce the cell proliferation rate by 50% (Table 2).

The non-monotonicity of the MTT test data may be explained by the following: we expect that at a low concentration of *DOX* and CaNP_7_–*DOX*, the efficiency of penetration of both the pure drug and the composition of nanoparticles affects cell survival. *DOX* in nanocomposites (CaNP_7_–*DOX*) most probably has better penetration and so is more effective at a concentration of 0.1 μM. With an increase in concentration, it has less effect, because nanostructures release *DOX* gradually and 0.5 μM is the maximum concentration of *DOX* in solution in the case of 100% release, which does not occur immediately, when free *DOX* is immediately added to this solution in full concentration. With a further increase in the concentration, *DOX* tends to be sorbed onto the surface (plate side) because of its hydrophobic properties [36,37,38]. In this case, *DOX* orubicin in the composition of nanocomposites is more effective. With a further increase in concentration, all these effects have less effect, since the concentration of *DOX*orubicin is high.

This experiment was primarily aimed at proving that the therapeutic properties of *DOX* are preserved when it is encapsulated in CaNP_7_. These CaNPs can serve as a delivery vehicle for a prolonged pH-dependent release of *DOX*, as depicted in the model experiments in Figure 7. Unfortunately, the MTT assay does not permit testing a pH-dependent release of *DOX* on the same cell type. Nonetheless, there is evidence in the literature that pH-sensitive materials manifesting high efficiency in vitro can be even more effective in vivo [4,12,13,14,15].

CaNP_7_–*DOX* turned out to be more cytotoxic than the free drug. Accordingly, we advanced a hypothesis that the greater reduction in the percentage of surviving cells is due to the gradual release of *DOX* from the CaNP_7_–*DOX* composite. In contrast, free *DOX* was added to the cells at a single time point at a final concentration and probably precipitated due to its hydrophobicity.

Thus, the CaNP_7_ nanocarrier proposed in this work was found to be nontoxic in vitro. The efficiency of inhibiting cancer cell growth by the CaNP_7_–*DOX* composite was not inferior to that of the free therapeutic agent. Nevertheless, due to the pH-dependent stability of the CaNP_7_ carrier, according to the kinetic profiles of the *DOX* release from CaNP_7_–*DOX* (Figure 8), a therapeutic efficacy is expected to be higher in vivo due to the selective dosed accumulation of the drug at a tumor site.

Our findings warrant further research into the application of CaCO_3_-based nanostructures in vivo.

## 4. Conclusions

Methods were developed for the preparation of monodisperse inorganic nanoparticles based on a CaCO_3_ (e.g., CaNP_7_) stable in suspension having sizes of 200 ± 20 nm. The synthesized nanoparticles were characterized by TEM and DLS. Due to their monodispersity and small size, the obtained materials were optimally suited for biomedical applications including intravenous drug delivery. Efficient conjugation of the obtained nanoparticles with an anticancer drug (*DOX*) was demonstrated, as was a pH-dependent profile of a *DOX* release from CaNP_7_–*DOX*. It is expected that the high efficiency of *DOX* encapsulation and the pH-dependent profile of its release will make it possible to reduce the amount of the administered drug in future in vivo experiments while increasing the effectiveness of the therapy; this is because the CaNPs provides a more efficient release of the drug from the nanoparticles in a site with lowered pH corresponding to a tumor macroenvironment. In the concentration range of 0.5–50.0 µg/mL, no toxicity of the CaNPs to the tested cell lines (HEK293, A549) was detectable. The properties of CaNPs should help to prevent unwanted accumulation of these nanomaterials in major organs such as the liver, heart, and kidneys, due to CaNP biodegradation. The effectiveness of cell growth inhibition by CaNP_7_–*DOX* turned out to be comparable to that of free *DOX* in the *DOX* concentration range of 0.1–10.0 µM. By means of the model drug *DOX* in the MTT assay, we proved that when our methodology is employed for constructing a pH-dependent transporter with a prolonged release ability, therapeutic efficacy of the drug is preserved. Our results indicate that the newly developed CaNPs hold great promise for further in vivo experiments.

## Figures and Tables

**Figure 1 nanomaterials-11-02794-f001:**
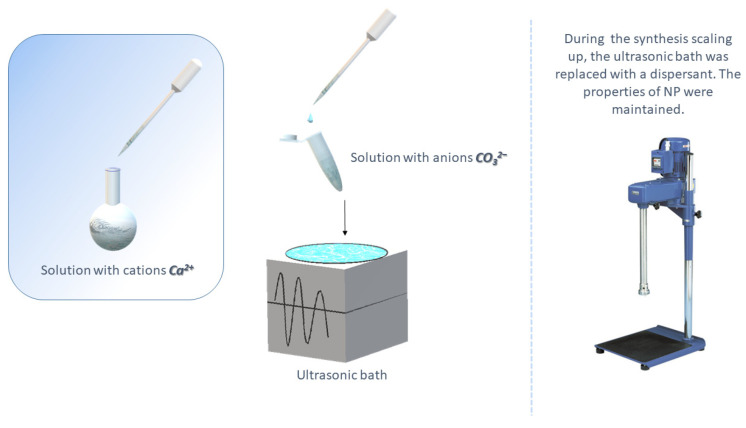
An outline of the fabrication of CaNPs.

**Figure 2 nanomaterials-11-02794-f002:**
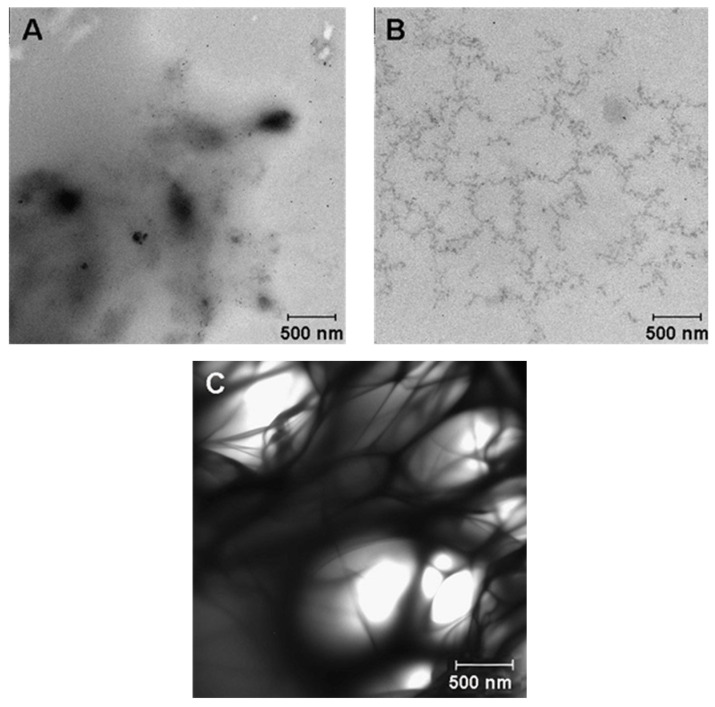
TEM images of Tween 20 (**A**), PEG-2000 (**B**), and the mixture of Tween 20 and PEG-2000 (**C**).

**Figure 3 nanomaterials-11-02794-f003:**
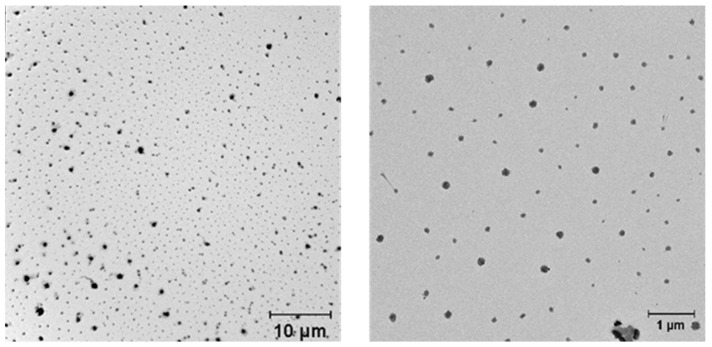
TEM images of the suspensions of the nanomaterial called CaNP_1_, which was produced in the presence of Tween 20 and PEG-2000.

**Figure 4 nanomaterials-11-02794-f004:**
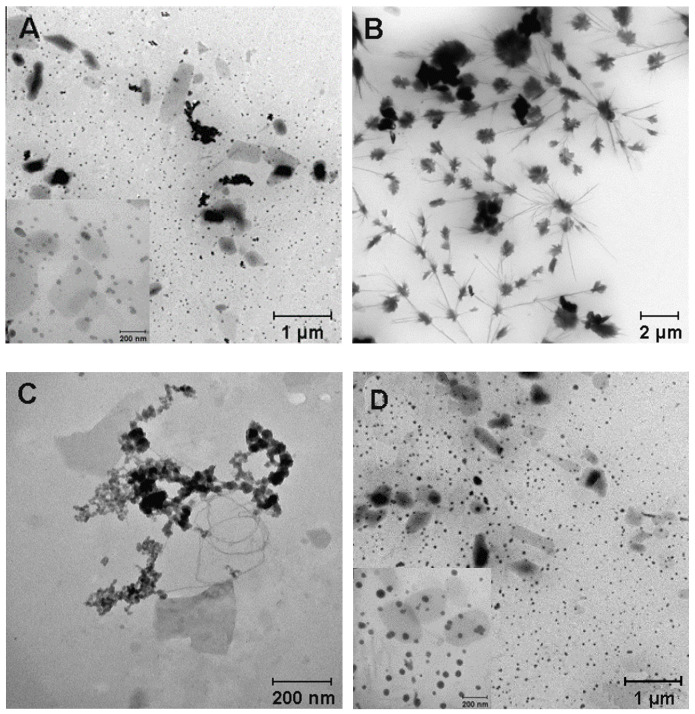
TEM micrographs of suspensions of CaNP_1_ (**A**), CaNP_2_ (**B**), CaNP_3_ (**C**), CaNP_4_ (**D**), CaNP_5_ (**E**), CaNP_6_ (**F**), CaNP_7_ (**G**), and CaNP_8_ (**H**).

**Figure 5 nanomaterials-11-02794-f005:**
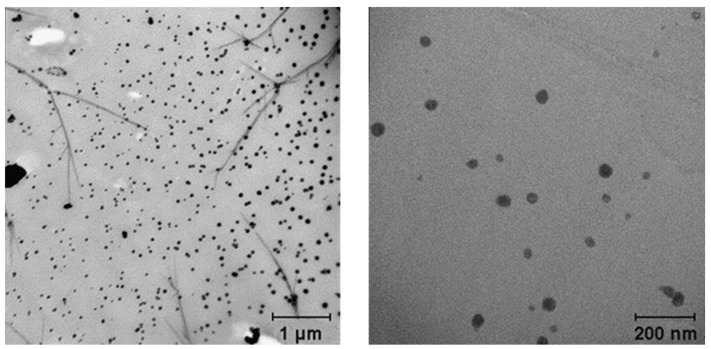
TEM images of suspensions of a CaNP_7_ preparation obtained in 30× reaction volume.

**Figure 6 nanomaterials-11-02794-f006:**
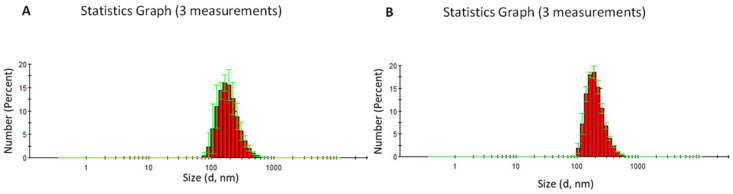
DLS size distribution of the CaNP_7_ nanoparticles. The particle size was determined by the DLS method to be 204 ± 8 nm, with a polydispersity index (PDI) of 0.14 ± 0.02 (**A**). DLS size distribution of CaNP_7_ nanoparticles after 90 days of storage. The particle size was determined by the DLS method to be 207 ± 4 nm, with a PDI of 0.11 ± 0.01 (**B**).

**Figure 7 nanomaterials-11-02794-f007:**
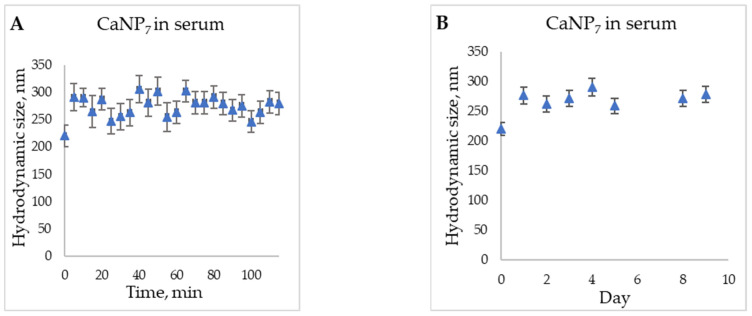
Stability of CaNP_7_ in 50% serum (FBS) according to DLS data during the first hours (**A**); within 10 days (**B**).

**Figure 8 nanomaterials-11-02794-f008:**
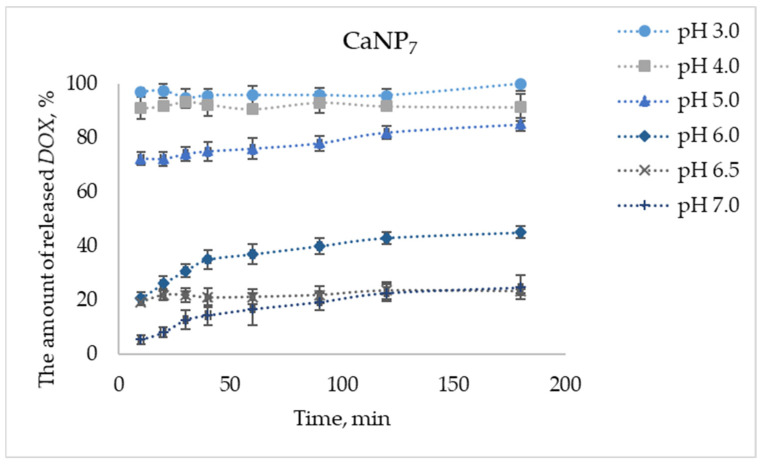
The proportion of *DOX* released from CaNP_7_–*DOX* with time at pH 3.0–7.0.

**Figure 9 nanomaterials-11-02794-f009:**
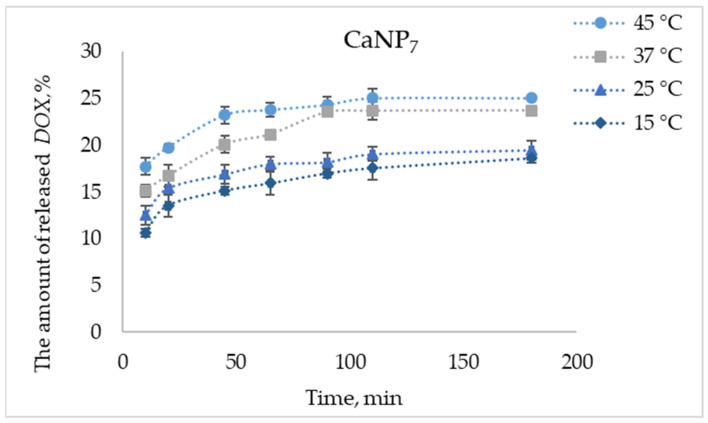
The proportion of released *DOX* (from CaNP_7_-*DOX*) as a function of time at pH 6.5 and different temperatures.

**Figure 10 nanomaterials-11-02794-f010:**
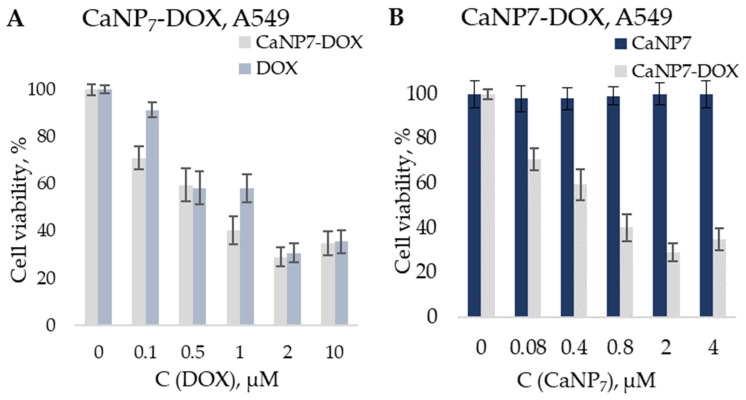
Cytotoxic activity of CaNP_7_, CaNP_7_–*DOX*, *DOX* on A549 cells. Cells were incubated with equimolar amounts of *DOX*, either soluble or loaded on nanoparticles (**A**) as well as with equivalent amounts of nanoparticles (**B**).

**Table 1 nanomaterials-11-02794-t001:** The influence of changing a single parameter in the CaNP_1_ fabrication workflow.

# (Figure 4 Panel)	Sample	Special Conditions	D, nm	PDI
1 (A)	CaNP_1_	-	339 ± 4	0.20 ± 0.01
2 (B)	CaNP_2_	Solvent: isopropanol	2637 ± 125	0.33 ± 0.05
3 (C)	CaNP_3_	MgCl_2_, 0.01 M	278 ± 4	0.15 ± 0.01
4 (D)	CaNP_4_	DMEM, 2 w.%	333 ± 2	0.26 ± 0.01
5 (E)	CaNP_5_	MgCl_2_, 0.01 M; DMEM, 2 w.%	326 ± 6	0.26 ± 0.01
6 (F)	CaNP_6_	MgCl_2_, 0.05 M; DMEM, 2 w.%	332 ± 1	0.19 ± 0.01
7 (G)	CaNP_7_	MgCl_2_, 0.01 M; DMEM, 10 w.%	249 ± 1	0.10 ± 0.01
8 (H)	CaNP_8_	MgCl_2_, 0.05 M; DMEM, 10 w.%	276 ± 4	0.10 ± 0.01

**Table 2 nanomaterials-11-02794-t002:** IC_50_ values.

Sample	IC_50_, µM
CaNP_7_–*DOX*	0.97 ± 0.04
*DOX*	2.41 ± 0.02

## Data Availability

Not applicable.

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
