# Peer review of "Designing pH-Dependent Systems Based on Nanoscale Calcium Carbonate for the Delivery of an Antitumor Drug"

_nanomaterials, 2021, doi:10.3390/nano11112794_

Round 1
Reviewer 1 Report
The manuscript “Designing pH-Dependent Systems Based on Nanoscale Calcium Carbonate for the Delivery of an Antitumor Drug” by Victoria Popova et al. reports on the preparation of calcium carbonate nanoparticles in different conditions of reagents aimed at finding the best combination in terms of different evaluation parameters (e.g. size, stability, …). The work is promising, but it requires to elucidate or complete the following points.
DOX release at normal pH of 7.0 is somehow high, thus resolving only partially the problem of the toxicity of free DOX. I suggest to take this into consideration and discussi it.
In the experiment reported in Fig 9, it is clear that increasing amounts of DOXO were used. What about the amounts of nanoparticles? Were they also increasing, since the same ratio nanoparticle:Doxo was used or different amounts of Doxo were coupled to the same amount of nanoparticles? I guess that the first option was followed. This must be clarified and requires a further experiment. I strongly suggest to add an experiments where increasing amounts of nanoparticles alone are tested in an MTT assay (see, for example, the paper Iafisco et al., Small 2013;9(22):3834-44, which I suggest to cyte).
It is not clear which cells were used for the experiment reported in Fig 9. This is not written both in the text and in the legend to Fig 9: indicate on which cell line the experiment reported in Fig 9 was done.
Line 453: I do not agree with the clain “this is because the nanocomposite may selectively accumulate at a site with lowered pH corresponding to a tumor macroenvironment.”. The lowered pH could contribute to a more efficient release of the drug from the nanoparticles, not necessarily to their accumulation in the tumor macroenvironment.
Since authors discuss the potential of their nanoparticles for in vivo experiments, they should be conscious that nanoparticles could be entrapped in some organs, e.g. liver, kidney, lung. They could add something.
Thus the manuscript can be accepted with revision, once the points raised have been answered or taken into consideration.
Author Response
Thank you very much for your time and valuable comments on our paper! All your comments were taken into account and the manuscript was edited according to them! Detailed answers to your review are presented in the attached file.

Reviewer 2 Report
- This manuscripts focus on improvement of the characterictics of encapsulation of Doxorubicin via CaNP. Shown as Figure 4, TEM revealed aggregation is still a big problem on other tested formulation except formulation CaNP7. However, the content didn't provide any stability test of desired formulation CaNP7, please provide the stability test or alternative of formulation CaNP7.
- For preparation of nanoparticles encapsulated with the modal drug for further clinical use, non-toxicity of nanoparticle is an important issue, however, the dose of DMEM is up to 10% (5 times of other tested formulation) to prepare this nanoparticle (formulation CaNP7), because DMEM is an toxic organic solvent, please describe the possible reason for increasing this organic solve or possible alternative method.
- Shown as Figure 9, the cytotoxicity on A549 cell line on both Doxurubicin and CaNP7-Doxurubicin with variable concentration seemed inconsistent and difficult to explain, why 0.1 uM and 1 uM doxurubicin-CaNps are more toxic than doxurubicin only, whereas, the toxicity of 0.5 uM doxurubicin-CaNp is similar to doxurubicin only ?
- Variable PH level change the release profile of Doxurubicin from the tested nanoparticle (CaNP7), and lower PH level increase the release of Doxurubicin from the tested nanoparticle (CaNP7). Do you have any data showing the effect of tumor penetration of these nanoparticles is better or other alternative method to approve these nanoparticles enhancing the anti-tumor effect in tumor environment?
Author Response
Thank you very much for your time and valuable comments on our paper! Detailed answers to your review are presented in the attached file.

Round 2
Reviewer 2 Report
- This manuscripts focus on improvement of the characterictics of encapsulation of Doxorubicin via CaNP. Tested formulation CaNP7 have few aggregation detected on the TEM and supplementary data showed PDI of CaNP7 is good and not significantly changed one month later. Please add this supplementary data on the formal manuscript.
- Figure 9 showed the comparison of dose escalating cytotoxicity between Dox and CaNP-Dox (A), and dose escalating cytoxicity between CaNP and CANP-DOx (B). It is still difficult to explain why 0.1 uM and 1 uM CaNps-Dox are more toxic than Dox only, whereas, the toxicity of 0.5 uM CaNp-Dox is similar to Dox only ? I think the possible reason is the A549 cell line for DOx is not the same generation compared with Dox-CaNP. Please delete Figure 9 or show as supplementary data and explain the reason of above inconsistent finding, and IC50 is enough for presenting the efficay of CaNP-Dox in this manuscript.
Author Response
Dear reviewer,
Thank you very much for your opinion, your time and valuable comments on our paper!
All your comments were taken into account and the manuscript was edited according to them ( see attached file)!
Thank you once more.
With the best regards,
On behalf of the authors,
Dmitry Pyshnyi and Elena Dmitrienko.

This manuscript is a resubmission of an earlier submission. The following is a list of the peer review reports and author responses from that submission.
Round 1
Reviewer 1 Report
In this manuscript authors describe the preparation of nanoscale particles based on calcium carbonate for drug delivery purposes. Specifically these nanoparticles are used for the delivery of doxorubicin. Though not very innovaitve or very original this study is accurate and basically well done and the ms well written and clear. THis ms may merit eventual publicaiton on Nanomaterials.
Porbably authors might expand the discussion of the obtained results and comment more extensivley their importance. There are a number of papers on doxorubicin and its delivery in the rencet literature that should be cited. Also the possible applications of these nanostructured materials in oncology could be commented in more depth. The English should be improved in several points.
Reviewer 2 Report
In this manuscript by Popova et. al, the authors demonstrate pH-dependent release of doxorubicin from CaNP carriers for inhibiting growth of cancer cells. The claims made by the authors are not consistent with the data presented in most parts of the manuscript, hence it is not suitable for publication in the journal. Below are some of my major concerns -:
- The difference in IC50 values of CaCo3-DOX and DOX is primarily due to the difference in cell viability at the concentration of 1uM. There is no difference in cell viability at any other doses. This might an experimental error. Also, CaCO3 should be used as a control.
- Cell Viability studies should also be performed at physiological pH of 6.5, to determine whether using CaNP as delivery vehicle has any effect on increasing the potency of DOX.
- In Figure 6, total amount of encapsulated drug should be calculated for each pH at t=0 and the resulting graph should be normalized w.r.t the initial concentration of the encapsulated drug.
- The stability of the CaNP should be tested in media containing 50% serum and reported using DLS.
- The author should include more appropriate references from groups that have done research on pH based drug delivery.
The results do not show any advantage of using CaNP as delivery vehicle over using the drug alone. The manuscript is not fit for publication in nanomaterials .